# Applications and Opportunities in Using Disulfides, Thiosulfinates, and Thiosulfonates as Antibacterials

**DOI:** 10.3390/ijms24108659

**Published:** 2023-05-12

**Authors:** Lindsay Blume, Timothy E. Long, Edward Turos

**Affiliations:** 1Department of Chemistry, University of South Florida, Tampa, FL 33620, USA; blumel@usf.edu; 2Department of Pharmaceutical Sciences, Marshall University, Huntington, WV 25755, USA; longt@marshall.edu

**Keywords:** antibiotic, antimicrobial resistance, sulfur, disulfide, thiosulfinate, thiosulfonate

## Abstract

Sulfur-containing molecules have a long history of bioactivity, especially as antibacterial agents in the fight against infectious pathogens. Organosulfur compounds from natural products have been used to treat infections throughout history. Many commercially available antibiotics also have sulfur-based moieties in their structural backbones. In the following review, we summarize sulfur-containing antibacterial compounds, focusing on disulfides, thiosulfinates, and thiosulfonates, and opportunities for future developments in the field.

## 1. Introduction

One of the most beneficial discoveries for public health are the drugs used to treat infectious diseases. Despite the success of antimicrobials in treating infections, there has been a drastic increase in the incidence of antimicrobial-resistant pathogens [1]. Diseases that were once easily treatable are becoming increasingly impervious to the usual treatments [2]. As antimicrobial resistance increases, few treatments have progressed to clinical trials in recent years. The increasing cost of progressing a new drug through the approval process is a barrier for pharmaceutical companies to invest in antimicrobial development. The vast majority of infectious diseases are cured with short-term therapies in immunocompetent patients to deter investment in their development [2,3].

Sulfur antibiotics encompass a vast array of bioactive organic compounds, with the main feature being a sulfur-sulfur bond. Many of these organosulfur compounds can be found in common plants such as garlic, onion, shallots, and chives and have antibacterial activity against a variety of pathogens. Additionally, several new sulfur-containing classes of antibiotics are used to treat multidrug-resistant pathogenic infections. This review will detail the pharmacological activity of disulfides, thiosulfinates, and thiosulfonates to showcase novel opportunities in antimicrobial drug development [4].

## 2. Early Studies on Naturally-Occurring Sulfur Compounds with Antimicrobial Properties

Extracts of *Allium* have been used since ancient times to treat bacterial infections and have been known to possess bioactive properties. A prolific amount of work over the last 60 years in Eric Block’s lab at the University of Albany has detailed the exquisite chemistry and metabolic products of *Allium* [5]. A beautiful summary of these investigations has been published, along with countless review articles, book chapters, books, and original research publications cited therein, all by Block and his colleagues. Figure 1 highlights some of the different metabolites from garlic that are of particular interest in this literature review.

All compounds encompassed in our discussion will in effect contain an electrophilic sulfur atom, presented as RS-X, that is expected to function as antibacterial agents by creating reactive oxygen species (ROS) and inhibiting metabolic pathways such as those for type II fatty acid biosynthesis. This general mode of action invokes a simple thio transfer from the sulfur electrophile to a reactive thiophilic substance within or prior to entering the cell, such as glutathione, coenzyme A, and cysteinyl enzymes (Figure 2). Transfer of the organosulfur moiety (RS-), in turn, creates a mixed disulfide capable of impairing metabolism and inducing oxidative stress by interfering with oxidant detoxification by glutathione or forming reactive oxygen species (ROS) that damage bacterial DNA [5]. Thus, the damage could be extensive, disrupting lipid synthesis and membrane formation, causing duplex DNA cleavage, or a host of other oxidative processes affecting the structural or metabolic integrity of the cell [6].

## 3. Disulfides

### 3.1. Ajoene

Disulfides encompass a vast array of bioactive organic compounds, with the main feature being a sulfur–sulfur bond. Natural disulfides such as ajoene (**1/2**) (Figure 3), found in garlic and other natural sources, are bioactive against a selection of Gram-positive bacterial strains [7] (Table 1). In particular, *Bacillus cereus*, *Bacillus subtilis*, *Mycobacterium smegmatis*, and *Streptomyces griseus* are reported to exhibit MIC values of 4 µg/mL. However, ajoene displayed no bioactivity against the Gram-negative microbes tested.

Mode of action studies wherein ajoene-infused media was combined with cysteine diminished ajoene concentrations and the antibacterial activity, suggesting that ajoene reacts intracellularly in bacteria with thiols to either cause antibacterial effects or reduce the efficacy of ajoene, depending on the thiol reacting with it [8].

### 3.2. Gliotoxin

The epipolythiodioxopiperazine (ETP) antibiotic gliotoxin (**3**) is a bridged disulfide (Figure 4) produced by many *Penicillium* and *Aspergillus* fungal species [9]. Gliotoxin (**3**) demonstrated MIC values of 4 µg/mL and 2 µg/mL against *S. aureus* ATCC 29213 and *S. aureus* ATCC 700699 strains, respectively [10].

Gliotoxin is thought to act through multiple pathways: (1) reduction to the bis-thiol followed by rapid air oxidation back to the disulfide, which results in the generation of superoxide and hydrogen peroxide that cause duplex DNA damage; and (2) sulfenylation of thiol groups in native proteins containing catalytically needed cysteine residues [10].

### 3.3. Aryl-Alkyl Disulfides

Revell et al. synthesized a variety of aryl-alkyl disulfide compounds (**4a**–**f**, **5a**–**d**, and **6a**–**d**) using aryl thiols coupled with alkyl thiosulfonates in a methanolic solution. Following work up, the aryl–alkyl disulfides were characterized and subjected to standard Kirby–Bauer disk diffusion testing against *Staphylococcus aureus*. Of the initial 8 disulfides synthesized, *para*-nitrophenyl methyl disulfide exhibited the highest antimicrobial activity, resulting in a zone of inhibition of 35 mm. Structural modifications of the most bioactive compound yielded the synthesis of 12 additional nitrophenyl alkyl disulfides, which were then subjected to Kirby–Bauer disk diffusion tests and serial dilution assays that gave their MIC values (Table 2) [11].

All the nitrophenyl disulfide compounds displayed activity against *S. aureus* and *B. anthracis*, with the most active series being the *para*-nitrophenyl and *meta*-nitrophenyl disulfides. Electron-deficient sulfur centers increased the nucleophilic scission of the aryl-alkyl disulfide bond. Comparing the *S*-alkylthio chains, an increase in length increased antibacterial activity, but rather moderately, whereas introducing branching increased activity drastically. The disulfide compounds also showed a somewhat higher efficacy against *S. aureus* than against methicillin-resistant *S. aureus*.

Further studies illustrated that the disulfide must be directly off the aryl ring in that the addition of a longer group as a linker (**7**,**8**) between the aryl ring and the disulfide effectively killed bioactivity. The corresponding *bis*(*p*-nitrophenyl) sulfide (**9**) was also synthesized and determined to be inactive against the bacterial strains tested; this confirmed unambiguously that the disulfide linkage is necessary for the antibacterial bioactivity (Figure 5).

Aryl–alkyl disulfides **7**–**9** assayed against purified *E. coli* β-ketoacyl-acyl carrier protein synthase III (FabH) in a buffered solution showed high inhibitory activity against the enzyme. These assays were completed in the absence of the additive coenzyme A, demonstrating the disulfide compounds are able to directly react with the thiophilic (thiol) functionality of the FabH protein to disrupt lipid biosynthesis in bacteria [11].

### 3.4. S,S′-bis(Heterosubstituted) Disulfides

In the Turos laboratory, a selection of low molecular weight *S*,*S*′-*bis*(heterosubstituted) disulfides were synthesized (Figure 6) to examine if different heteroatoms attached to the sulfurs affected antimicrobial activity in some way. To make these derivatives, sulfur monochloride was added to either an amine, alcohol, or a thiol with triethylamine at −20 °C. The pure *S*,*S*′-*bis*(heterosubstituted) disulfide products were obtained after column chromatography in high yields. Antibacterial testing and growth viability assays were employed against selected bacteria and also against the fungus *Candida albicans* [12].

Antibacterial testing of the synthesized compounds revealed that *S*,*S*′-*bis*(alkoxy) disulfides gave the best activity against methicillin-resistant *Staphylococcus aureus* (MRSA). The most potent analog, *S*,*S*′-*bis*(isopropoxy) disulfide (**10**), showed MIC values of 0.5 µg/mL against MRSA. Incubation of MRSA with **10** followed previously reported data that the antibiotic activity was canceled out in the presence of a nucleophilic sulfur species such as added glutathione. This suggests the formation of mixed alkyl-CoA disulfides, which inhibit the FabH protein and block fatty acid biosynthesis in bacteria. Further antibacterial testing showed robust activity against *Francisella tularensis*, the causative agent for the disease tularemia, with MIC values of 1 µg/mL for *S*,*S*-*bis*(phenylamino) disulfide (**11**) [12].

### 3.5. Pyridyl Disulfides

Pyridyl alkyl disulfides (**12**) are electrophilic sulfur species made synthetically in the Long lab and have been shown to be capable of mimicking natural allicin obtained from garlic in their bioactivity (Figure 7). The skeletal structure contains an electrophilic sulfur atom, making it vulnerable to thiol-disulfide exchange reactions with cellular thiols such as glutathione, coenzyme A, or mycothiol. Gram-positive species, including vancomycin-intermediate and vancomycin-resistant *S. aureus* (VISA and VRSA), exhibited susceptibility toward analogs with *S*-alkyl chains of 7 to 9 carbons in length. Checkboard assays revealed that compounds were synergistic with vancomycin against VRSA, induced dispersal of *S. aureus* biofilms, and had low cytotoxicity. Disulfide **13** was found to have the highest bioactivity against various multidrug-resistant *Staphylococcus aureus* strains [13].

### 3.6. Thiuram and Disulfiram

Thiuram (**14**) is a disulfide used as a fungicide to keep seeds and crops fungi-free and also as an animal repellant (Figure 8). Disulfiram (**15**), the ethyl analog commonly known as Antabuse, is used clinically to treat chronic alcoholism as it inhibits acetaldehyde dehydrogenase and increases the concentrations of acetaldehyde in the blood. This in turn enhances the feeling of hangover symptoms as a deterrent against ethanol consumption. Several articles have been published to repurpose oral and topical formulations of disulfiram as an anti-infective [14,15]. Owing to its chemical reactivity with cellular thiols, the antimicrobial spectrum of disulfiram encompasses a range of bacteria, fungi, viruses, protozoa, and helminths. Recent reports have described, for instance, activity against *M. tuberculosis* [16], *F. tularensis* [17], *Borrelia burgdorferi* [18], and Gram-positive bacteria [19], including MRSA [20].

The Long lab tested thiuram and disulfiram against 30 strains of vancomycin-susceptible, vancomycin-intermediate, and vancomycin-resistant *S. aureus*. MIC_90_ values ranged from 4–16 µg/mL. A checkerboard assay was performed to determine if disulfiram and vancomycin together exerted a synergistic effect on bioactivity. The assay indeed showed disulfiram potentiated vancomycin susceptibility with vancomycin-resistant *S. aureus* strains. Current work is focused on developing oral disulfiram as an adjunct therapy for systemic infections caused by vancomycin-intermediate *S. aureus* (VISA) [14].

### 3.7. Unsymmetrical Monoterpenylheteroaryl Disulfides

A series of 16 unsymmetrical monoterpenylheteroaryl disulfides were likewise synthesized for microbiological evaluation (Figure 9) [21]. These compounds were subjected to MIC assay against several Gram-positive bacterial species including methicillin-susceptible *S. aureus* and methicillin-resistant *S. aureus*, as well as against Gram-negative *Pseudomonas aeruginosa* and a *Candida albicans* yeast strain. Compound **16** reportedly exhibited the most bioactivity at 16 µg/mL against both Gram-positive and Gram-negative bacteria. Compounds **17** and **18** also exhibited moderate activity, but only against MSSA. These are optically pure compounds, but the effect of chirality was not examined in this reported study.

Despite the mild bioactivity of several monoterpenylheteroaryl disulfides, cytotoxicity studies showed selectivity indices between 1 and 12, with the majority under 5. Using these compounds as systemic agents for antibacterial treatments would be unlikely due to their moderate to high cytotoxicities; however, further research may indicate their possible use as topical antibacterials [21].

### 3.8. Pyridine-N-oxide Disulfides

Three pyridine-*N*-oxide disulfides (**19**–**21**) were extracted and characterized from *Allium stipitatum*, more commonly known as Persian shallot (Figure 10). They were subjected to MIC assays against several Gram-positive bacteria, including multiple strains of *Mycobacterium* and *Staphylococcus aureus*. Compounds **19**, **20**, and **21** exhibited potent antibacterial activity, with the lowest MIC value of 0.5 µg/mL being against the EMRSA-15 *S. aureus* strain.

Based on the structures of the three natural products, an additional five heteroaromatic methyl disulfides (**22**–**26**) were synthesized for comparison. All five of these also had antibacterial activity against the tested pathogens, mirroring the trend of the natural products, wherein the best activity was against the EMRSA-15 *S. aureus* strain. In vivo antibacterial testing of **19** in mice infected with *Mycobacterium tuberculosis* was undertaken, showing no clinical difference in bacterial numbers between untreated controls and those treated with compound **19**. Despite low in vivo antibacterial activity, cytotoxicity studies against carcinoma and adenocarcinoma lines suggest potential as an antitumor cancer treatment [22].

### 3.9. Aromatic and Heterocyclic Methyl Disulfides

A collection of disulfide analogs from *Allium stipitatum*, more commonly known as Persian shallot, were synthesized based on previously reported pyridine-*N*-oxide disulfides (Figure 11) [22]. Mild antibacterial activity was seen against several bacterial species with MIC values of 16–32 µg/mL. The most bioactive compound was determined to be compound **27**, with a MIC value of 4 µg/mL against *Mycobacterium tuberculosis*.

In addition to MIC testing, efflux pump inhibition was also examined. In some multidrug-resistant Gram-negative bacteria, a common mechanism of resistance is the specialization of efflux pumps to remove antibiotics against a concentration gradient from inside the cell to outside the cell. Ethidium bromide was used as an efflux pump substrate; when a compound interfered with an efflux pump, the ethidium bromide concentration in the cell increased and was detected with fluorescence emissions. Compounds **27** and **28** showed this fluorescence with relative fluorescent units (RFU) of 45 and 41, respectively, compared to the positive control, verapamil, which produced an RFU of 53 [23].

### 3.10. Disulfide-Containing Vancomycin Derivatives

A series of vancomycin derivatives were synthesized containing a disulfide or thiol moiety to assess activity against multidrug-resistant bacteria (Figure 12). The compounds were active against both Gram-positive multidrug-resistant bacteria and, less expectedly, the Gram-negative bacterium *Moraxella catarrhalis*. The MIC data show a clear increase in antibacterial activity compared to vancomycin as a standard. Against the Gram-negative bacterium *Moraxella catarrhalis*, compound **30o** had an increase in activity from standard vancomycin with a MIC value above 32 µg/mL to 0.5 µg/mL for compound **30f**.

Antibacterial activity against biofilm formation was also determined, showing no significant improvement against multidrug-resistant *Staphylococcus* strains. However, against vancomycin-resistant *Enterococcus faecalis*, an increase in MBIC values was displayed. In particular, compounds **30e** and **30f** showed biofilm inhibition compared to standard vancomycin [24].

### 3.11. Calicheamicins

Calicheamicins (**31**, and esperamicins) are a group of enediyne antitumor antibiotics first isolated by the Lederle Labs over 30 years ago from the Gram-positive bacterium *Micromonospora echinospora* [25].

Mode of action studies determined a complex antibacterial and antitumor mechanism (Figure 13). These compounds bind in the minor groove of DNA and undergo activation by a nucleophilic attack from glutathione at the central sulfur atom of the trisulfide (**31**). Once the sulfur-sulfur bond is broken, the resulting highly nucleophilic allylic thiolate anion undergoes an intramolecular Michael addition to the β-carbon of the enone. The generation of the five-membered sulfur ring changes the restricted geometry of the enediyne moiety enough to trigger a Bergman cyclization. The resulting radicals quickly abstract protons from the deoxyribose rings from the backbone of duplex DNA, leading to scission of the strand and breakdown of the DNA [26].

Antibacterial activity was determined by standard broth dilutions in Mueller–Hinton medium. Activity was seen across all species tested, including Gram-positive bacteria *Staphylococcus aureus*, *Enterococcus* sp., *Bacillus subtilis*, and Gram-negative bacteria *Escherichia coli, Klebsiella pneumoniae*, *Enterobacter* sp., *Serratia* sp., *Citrobacter* sp., *Acinetobacter* sp., and *Pseudomonas aeruginosa*, with antibacterial activity under 1 µg/mL [27].

### 3.12. Marine Disulfide Natural Products

A disulfide-containing compound, citorellamine (**32**), was isolated from the tunicate *Polycitorella mariae* (Figure 14). First misinterpreted as a sulfide, structure revision determined the disulfide functionality. Citorellamine showed high activity against several bacterial strains, including *Staphylococcus aureus*, *Bacillus subtilis*, and *Escherichia coli* [28].

Several compounds with antibacterial activity were isolated as natural products from marine sponges. Three of these were bromotyrosine disulfide derivatives (Figure 15) and were isolated from species of *Psammapliysill* and *Thorectopsamma xana* sponges [29]. Upon antibacterial testing, psammaplin A (**33**) and D (**34**), and the dimer bisaprasin (**35**) each inhibited the growth of *Staphylococcus aureus* and *Bacillus subtilis*. In addition to the Gram-positive bacteria above, psammaplin D showed antifungal activity against *Trichophyton mentagraphytes* (ringworm). Quantitative MIC values were not reported [29,30].

## 4. Thiosulfinates

### 4.1. Thiosulfinates from Garlic Extract

The most common thiosulfinates cited for antibacterial applications are compounds extracted from plants from the genus *Allium*, which include garlic, leeks, shallots, chives, onions, etc. Allicin is one of the most well-known compounds extracted from garlic. Belonging to the thiosulfinate family, allicin by itself does not exist in the bulb. However, upon mechanical crushing or cutting, the enzyme alliinase is activated, and the precursor alliin (**36**) is broken down into allylsulfenic acid, which then dimerizes with dehydration to form allicin (**38**) (Figure 16) [31].

The antibacterial activity of allicin (**38**) has been extensively studied, and its antibacterial properties have been known since the 1940s. Allicin has been shown to have antibacterial bioactivity against both Gram-positive and Gram-negative bacteria. MIC values against Gram-positive bacteria, including *Bacillus cereus*, *B. subtilis*, *Staphylococcus aureus,* and *Micrococcus luteus,* ranged from 5–10 µg/mL. Gram-negative strains except for *Pseudomonas aeruginosa* (MIC > 100 µg/mL) ranged from 15–30 µg/mL [7].

Raw garlic was crushed, homogenized with water, and filtered. The solid residue was subjected to steam distillation, and the resulting extract was diluted ten times with dichloromethane. HPLC was used to separate the various components of this extract to yield allicin as well as previously unreported thiosulfinate compounds stable enough to determine their antimicrobial MICs (Figure 17). The compounds were tested against Gram-positive and Gram-negative bacteria and against two yeast species. Allicin (**38**) in general showed the highest bioactivity, with MIC values ranging from 5–10 μg/mL against Gram-positive bacteria. The activity of all thiosulfinates (**39**–**41**) against Gram-negative bacteria was modest, with MIC values of 15–100 μg/mL [32].

In addition, *S*-methyl methanethiosulfinate (**42**) and *S*-propyl propanethiosulfinate (**43**) and allicin (**38**) were tested against multidrug-resistant strains of bacteria cultured from the lungs of cystic fibrosis patients (Figure 18). Three strains of multidrug-resistant Gram-negative bacteria, *Achromobacter ruhlandii*, *Burkholderia cenocepacia*, and *Pseudomonas aeruginosa*, were treated with the compounds pictured. MIC assays showed the strongest antibacterial activity from **40** and **36** (allicin) against *B. cenocepacia* at 30 μg/mL [33].

### 4.2. Synthetic Allicin Derivatives 

In a study published from Aachen University in Germany, allicin thiosulfinate derivatives were synthesized and tested against several strains of Gram-positive bacteria and yeast [34]. As is shown in Table 3, the thiosulfinate compounds had greater bioactivity against *S. cerevisiae* (baker’s yeast) versus the Gram-positive bacteria. Diallylthiosulfinate (**38**) and *S*-propyl propanethiosulfinate (**43**) had the best activity with MFC concentrations of 2 µg/mL (Table 3).

Mild bioactivity was observed, most specifically with gas-phase application of the compounds. On a Petri dish lid, a 20 µL drop of 80 mM compound solution was dropped. The base of the Petri dish was filled with bacteria-seeded medium and then suspended above the lid. This allowed for no direct contact between the test solution and the agar itself. All of the compounds except for benzyl benzenethiosulfinate showed zones of inhibition above where the droplet was placed. Plate inhibition zone assays followed the above trend, with *S*-methyl methanethiosulfinate demonstrating the highest activity against *Pseudomonas syringae* pv. *phaseolicola* 4612 Ps4612 and *Saccharomyces cerevisiae* BY4742 Sc (baker’s yeast) [35].

### 4.3. Leinamycin

Leinamycin (**44**) is a natural product produced by *Streptomyces atroolivaceus* and contains a cyclic thiosulfinate group. In cellular environments, leinamycin reacts with thiols to form an episulfonium cation (Figure 19). This three-membered, positively charged sulfur ring is susceptible to nucleophilic attack, most specifically by nucleophilic guanine residues in bacterial duplex DNA. The mechanism for cellular replication is therefore shut down in the presence of leinamycin [36]. Undergoing antibacterial testing, Leinamycin showed potent antibacterial activity against *Bacillus subtilus* with a MIC value of 0.03 μg/mL [36].

In addition to leinamycin’s antibacterial activity, new compounds related to leinamycin are being bioengineered for biotesting. Human-directed manipulation of gene clusters of leinamycin-producing actinomycetes strains is being developed with the introduction of diversity in the core chemical structure through modification of polyketide synthase enzymes [37].

## 5. Thiosulfonates

Further oxidation of the sulfur atom in thiosulfinates affords thiosulfonates (Figure 20).

Two compounds extracted from garlic, *S*-propyl propanethiosulfinate (**43**) and *S*-propyl propanethiosulfonate (**46**), were tested against a variety of Gram-positive and Gram-negative bacteria (Figure 21). Both compounds had minimal activity against Gram-negative strains. Against Gram-positive methicillin-resistant *Staphylococcus aureus*, *Enterococcus faecalis*, and *Streptococcus agalactiae*, *S*-propyl propanethiosulfonate (**46**) showed higher activity with MIC values of 4–8 μg/mL [38].

Continued studies from the University of Granada-ibs in Spain replicated the data reported above and expanded upon it to determine the antimicrobial effects of *S*-propyl propanethiosulfinate (**43**) and *S*-propyl propanethiosulfonate (**46**) via the gas phase. Inhibition assays against all the Gram-positive and Gram-negative bacteria tested (sans the multidrug-resistant *Pseudomonas aeruginosa*) showed larger inhibition zones with *S*-propyl propanethiosulfonate (**46**) than with *S*-propyl propanethiosulfinate (**43**) [39]. Using a mixture of compounds **43** and **46** as a dietary adjunct for gilthead sea bream juveniles also decreased antibacterial and antiparasitic infections in the fish population. This increased their survival rate to 91.1% from the control group, which has an overall survival rate of 66.7% [40].

### 5.1. Quinoline Thiosulfonate Derivatives

Quinolines are often used as pesticides in agriculture; oxidized sulfur compounds are well known for their antibacterial activity. Based upon this rationale, a series of seven quinoline thiosulfonate compounds (**47a**–**g**) were synthesized (Figure 22). Preliminary data showed bioactivity against Gram-negative *Escherichia coli* and the fungus *Candida albicans*. Unfortunately, antimicrobial MIC data and details about the possible mechanism of action were not described, which perhaps warrants further testing of **47a**–**g** against additional pathogens [41].

### 5.2. Topical Thiosulfonate and Biosurfactant Combination

*S*-ethyl para-aminobenzenethiosulfonate (**48**) was synthesized and combined with the biosurfactant rhamnolipid biocomplex PS (**49**) derived from *Pseudomonas* sp. (Figure 23) [42].

The mixture was tested along with approved medicinal agents against the fungus *Candida albicans* and the bacteria *Staphylococcus aureus* and *Escherichia coli*. Disk diffusion assay showed that *S*-ethyl para-aminobenzenethiosulfonate (**48**) co-mixed with the biosurfactant (**49**) inhibited the growth of *Candida albicans* with a zone of inhibition of 32 mm. Bacterial growth was inhibited for *Staphylococcus aureus* with a zone of inhibition of 30 mm and for *Escherichia coli* with a zone of inhibition of 23 mm. These zones of growth inhibition indicate that the ointments of **48** and **49** have greater antibacterial activity than currently approved ointment treatments (Table 4). In addition to stronger bactericidal activity, the thiosulfonate/biosurfactant ointment is equal to or cheaper than the other approved treatments, with the exception of clotrimazole.

### 5.3. Preliminary Data for New Antibacterial Thiosulfonate Candidates

Most recently, in our laboratory, synthetic thiosulfonates (Figure 24) were examined for antibacterial screening [43]. Of the 23 *S*-alkyl thiosulfonates prepared, 20 of them were synthesized by the reaction of a symmetrical disulfide with the sulfinic acid salt [44]. The other three thiosulfonates, compounds **70**–**72**, were synthesized through the oxidation of a symmetrical disulfide using mCPBA as an oxidizing agent [45].

The *S*-alkyl thiosulfonates **50**–**72** underwent antibacterial testing against the bacterial pathogens shown in Table 5. The strains tested encompassed the ESKAPE pathogens (multidrug-resistant superbugs): methicillin-resistant *Staphylococcus aureus* COL, vancomycin-intermediate *Staphylococcus aureus* MU50, vancomycin-resistant *Staphylococcus aureus* VRSA-1, vancomycin-resistant *Enterococcus faecium* HF50104, *Klebsiella pneumoniae* 700603, *Acinetobacter baumanni* 5075-UW, *Pseudomonas aeruginosa* 15442, and *Enterobacter cloacae* 13047.

The following 11 *S*-alkyl thiosulfonate esters were demonstrated to show no inhibition of bacterial growth against any of the strains tested: **50**, **51**, **52**, **54**, **55**, **56**, **57**, **59**, **60**, **61**, and **72**. Seven *S*-alkyl methanethiosulfonates, *S*-alkyl ethanethiosulfonates, and *S*-alkyl propanethiosulfonates (except for *S*-propyl ethanethiosulfonate) demonstrated mild bioactivity.

The most potent of these compounds was *S*-propyl propanethiosulfonate (**70**), with a MIC value of 4 μg/mL against vancomycin-intermediate *Staphylococcus aureus* and a MIC value of 2 μg/mL against vancomycin-resistant *Staphylococcus aureus.* Compound *S*-ethyl propanethiosulfonate (**62**) showed mild bioactivity against the *S. aureus* strains tested with MIC values of 16 μg/mL.

Branched thiosulfonates containing the *sec*-butyl functionality (**53**, **58**, **63**, **68**, and **71**) showed even higher activity. The least bioactive analog among these was *S-sec*-butyl methanethiosulfonate (**53**), and the compound exhibiting the strongest bioactivity was the symmetrical *S-sec-*butyl *sec*-butanethiosulfonate (**71**).

*S*-Propyl propanethiosulfonate (**70**) and *S-sec-*butyl *sec*-butanethiosulfonate (**71**) displayed the strongest activity against vancomycin-intermediate *Staphylococcus aureus* and vancomycin-resistant *Staphylococcus aureus,* with a MIC value of 4 μg/mL and 2 μg/mL, respectively. Against the trend, *S*-isopropyl isopropanethiosulfonate showed no activity in triplicate assays against the bacteria tested.

Some branched *S*-alkyl thiosulfonates containing the *iso*-propyl group (**64** and **69**) also displayed strong bioactivity against the pathogenic bacterial strains tested. The most bioactive among this group against the Gram-positive *Staphylococcus aureus* strains was *S-iso*-propyl ethanethiosulfonate (**64**), although *S-iso*-propyl benzenethiosulfonate (**69**) showed higher activity against vancomycin-resistant *Enterococcus faecium*. These are shown in Table 6 for clarity.

The results of the assays follow previously reported data from the Turos lab on aryl-alkyl disulfides: that *S*-alkyl chain branching increases bioactivity, particularly against Gram-positive *S. aureus* strains. This suggests the bioactive mechanism of these *S*-alkyl thiosulfonates mimics the mode of action of allicin and previously reported disulfide species, where enzymes containing thiol-groups are inactivated upon reaction with these electrophilic sulfur species, including those in bacterial fatty acid biosynthesis [35].

## 6. Conclusions

Antibacterial extracts and compounds containing sulfur have been utilized for centuries. Many reported organosulfur compounds have been shown to be effective antibacterials or antibiotic adjuvants against multidrug-resistant bacteria, including MRSA. The unique mechanism of action of these organosulfur species as an antimetabolite and inducer of oxidative stress to inhibit growth remains an attractive yet relatively unexplored opportunity for antibacterial development.

Among these are a new series of *S*-alkyl thiosulfonates that were synthesized and described in this review. Several of the thiosulfonate compounds exhibited strong bioactivity against VISA (vancomycin-intermediate *Staphylococcus aureus*) and VRSA (vancomycin-resistant *Staphylococcus aureus*). These promising preliminary results warrant further investigation.

As the repertoire of antibacterial compounds continues to lose efficacy, imaginative designs targeting less explored bactericidal pathways are necessary. The molecules described in this review and the preliminary data reported above are a hopeful step in the right direction for controlling or even eradicating pathogenic bacteria based on the thiolation of key cellular functionality such as cysteinyl proteins and cellular thiols that so far have not been fully appreciated as potential new targets for drug discovery and development.

Disulfides, thiosulfinates, and thiosulfonates are a unique family of sulfur-containing compounds with often intriguing biological activity, including antibacterial properties. Those described in the scientific and patent literature have been highlighted in this summary. Their biological effects are directly related to their ability to serve as reductants and sulfur-transfer reagents inside cells, including human and bacterial ones. There are many things still left to explore for each of these compounds in order to better understand their metabolic fates and also their impact on enzymes and other cellular components relating to the biological effects the compounds may have in vitro versus in vivo. Moreover, chemists can create new hybrid structures to embody the thio-transfer capabilities of other classes of antibiotics, opening the spectrum of bioactivity and modes of action. This could be of vital interest in dealing with the ever-growing problems of multi-drug bacterial infections, such as MRSA and VISA. However, sustained funding and investment in research and development would be required, both in the pharmaceutical and academic sectors, and at the present time, this remains the most difficult and detrimental aspect of drug discovery and development for bacterial applications. Antibacterial resistance will undoubtedly continue to be an increasing concern in our medical care and in the community, in the upcoming years, so perhaps this area may be of interest to those looking for innovative solutions to create effective new antibacterials.

## Figures and Tables

**Figure 1 ijms-24-08659-f001:**
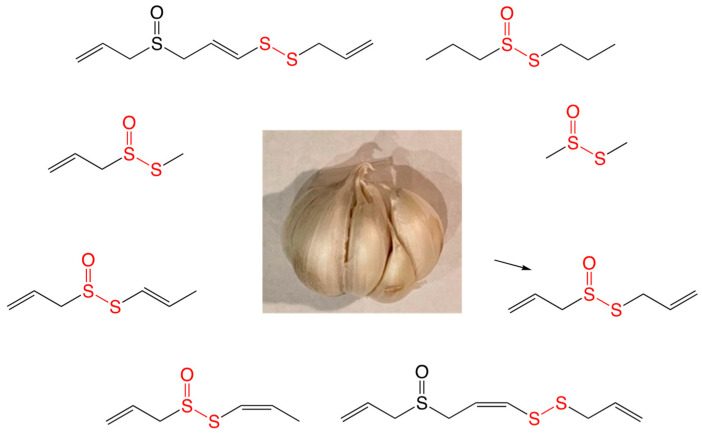
Antimicrobial organosulfur compounds found in *Allium sativum*.

**Figure 2 ijms-24-08659-f002:**
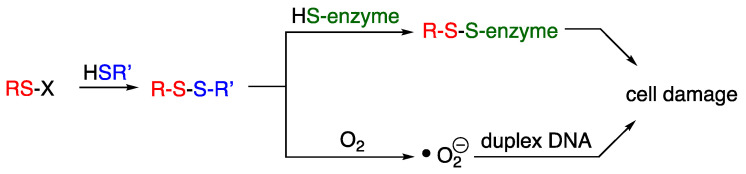
Thio transfer via nucleophilic displacement creates a new mixed disulfide species that leads to cell damage.

**Figure 3 ijms-24-08659-f003:**
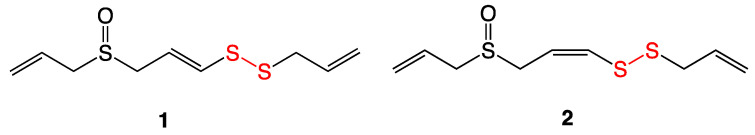
*E*/*Z* stereoisomers of ajoene (**1** and **2**).

**Figure 4 ijms-24-08659-f004:**
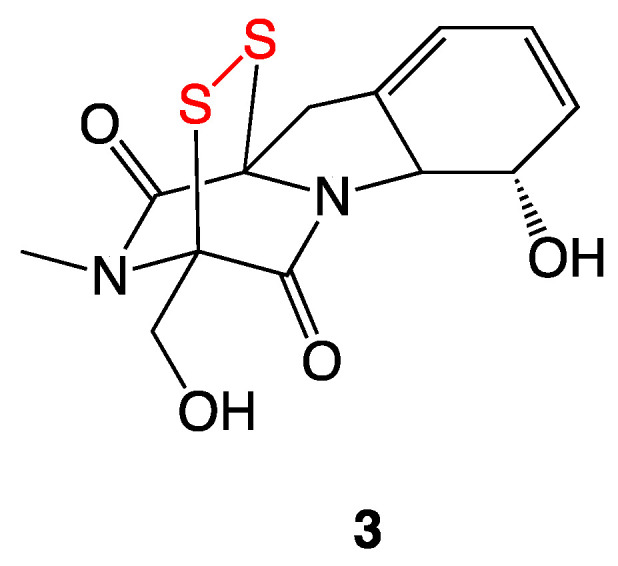
Chemical structure of gliotoxin (**3**) with disulfide moiety highlighted in red.

**Figure 5 ijms-24-08659-f005:**
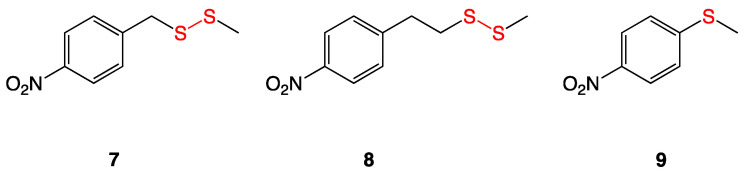
Sulfur analogs with no antibacterial activity.

**Figure 6 ijms-24-08659-f006:**
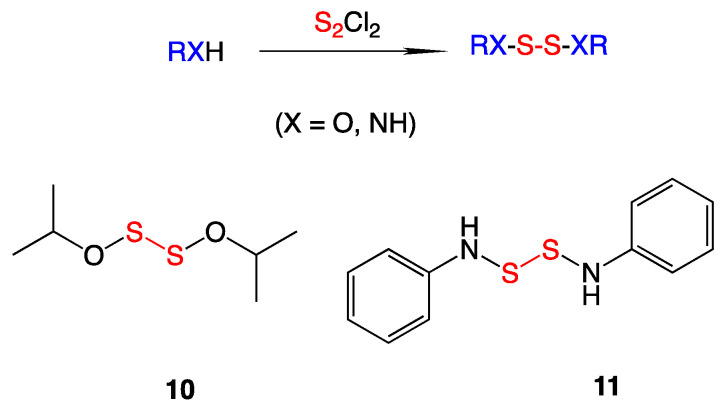
Reaction scheme and most bioactive heterosubstituted disulfides against *S. aureus* and *Francisella* species.

**Figure 7 ijms-24-08659-f007:**
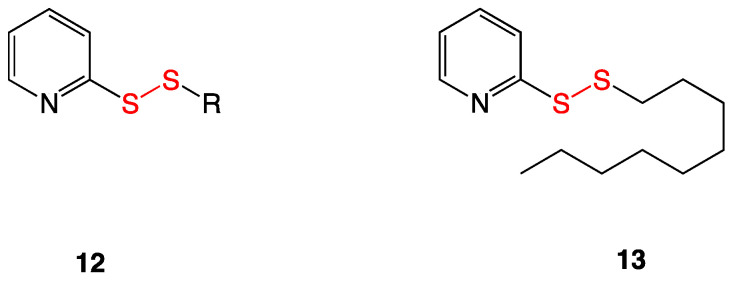
Pyridyl alkyl disulfide structure (**12**) and the most bioactive pyridyl nonyl disulfide (**13**) [12].

**Figure 8 ijms-24-08659-f008:**
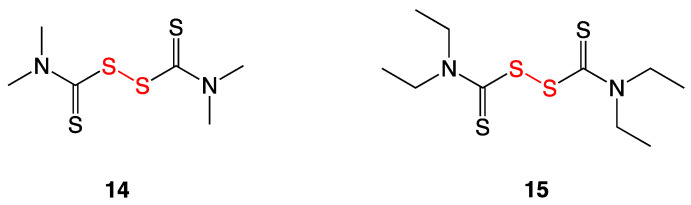
Structures of thiram (**14**) and disulfiram (**15**).

**Figure 9 ijms-24-08659-f009:**
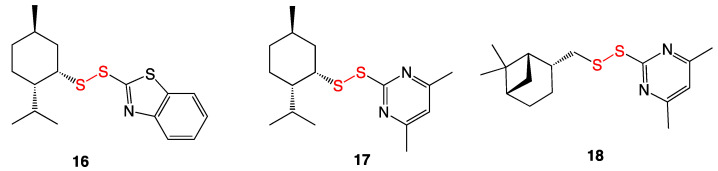
Synthesized unsymmetrical monoterpenylheteroaryl disulfides.

**Figure 10 ijms-24-08659-f010:**
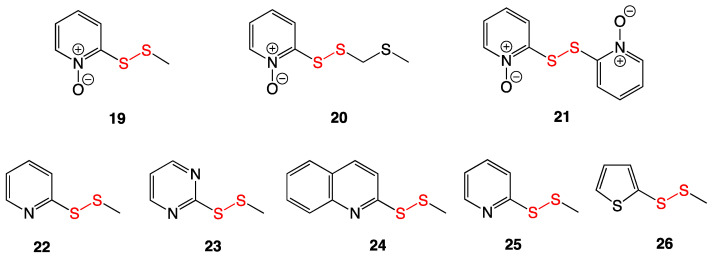
Pyridine-*N*-oxide disulfides (**19**–**21**) extracted from *Allium stipitatum* and synthetic analogs **22**–**26**.

**Figure 11 ijms-24-08659-f011:**
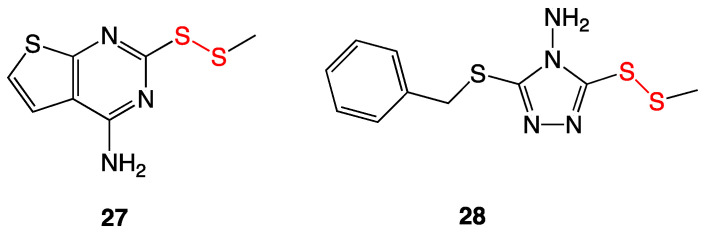
Bioactive aromatic and heterocyclic methyl disulfides from Persian shallot.

**Figure 12 ijms-24-08659-f012:**
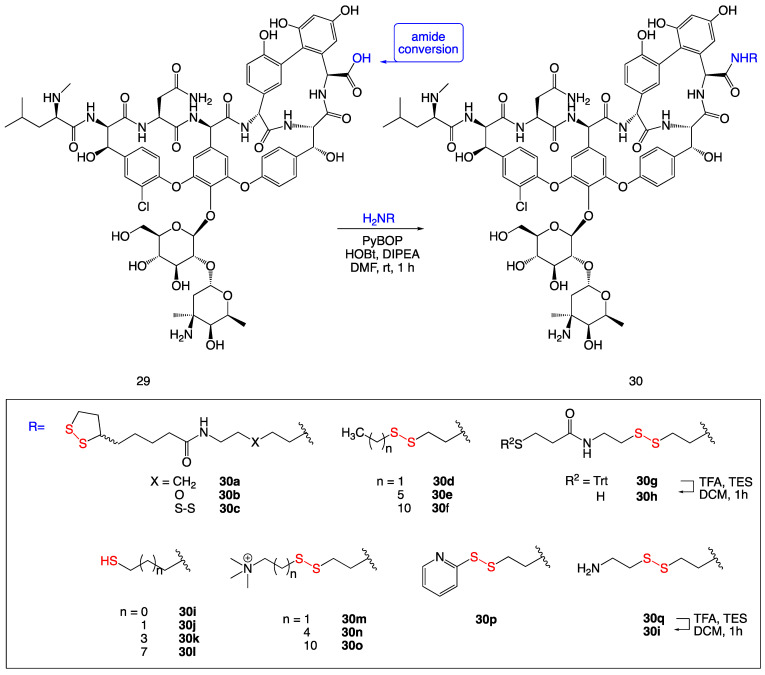
Thiol and disulfide-contained vancomycin derivatives.

**Figure 13 ijms-24-08659-f013:**
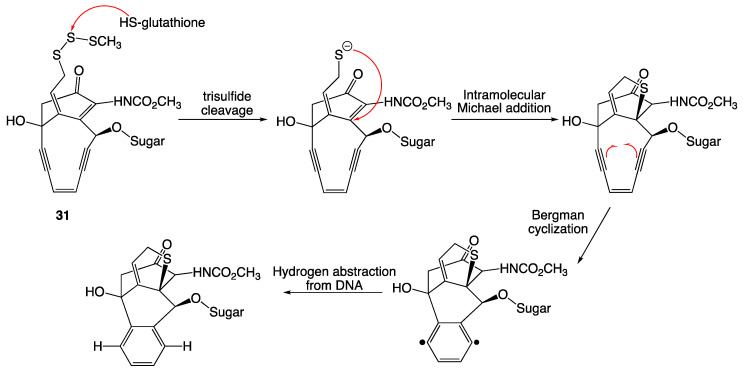
Mechanism of action for calicheamicin.

**Figure 14 ijms-24-08659-f014:**
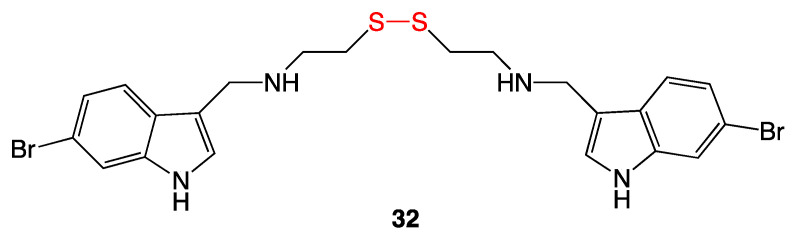
Citorellamine (**32**).

**Figure 15 ijms-24-08659-f015:**
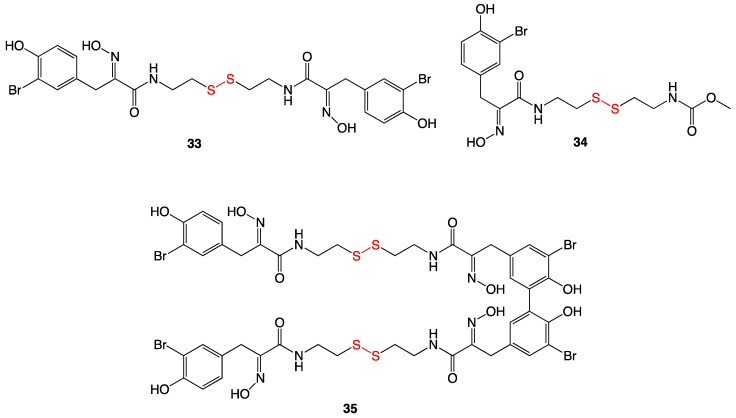
Isolated disulfide marine natural products.

**Figure 16 ijms-24-08659-f016:**
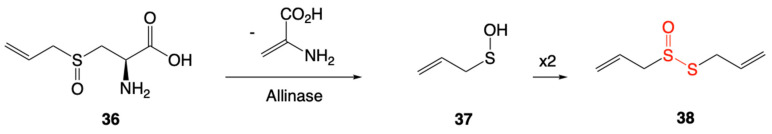
Biosynthesis of allicin (**38**) from alliin (**36**) [4].

**Figure 17 ijms-24-08659-f017:**
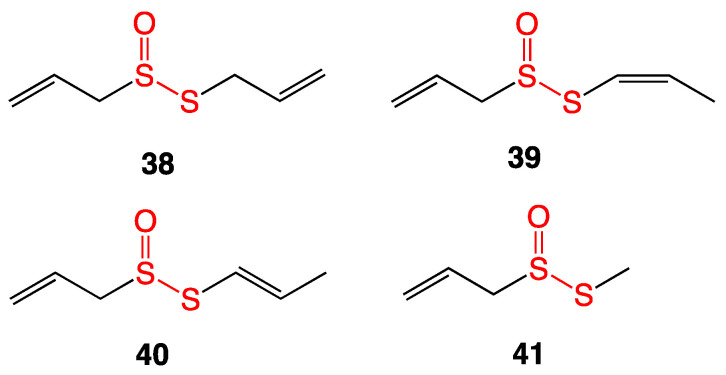
Extracted thiosulfinate compounds from macerated garlic.

**Figure 18 ijms-24-08659-f018:**
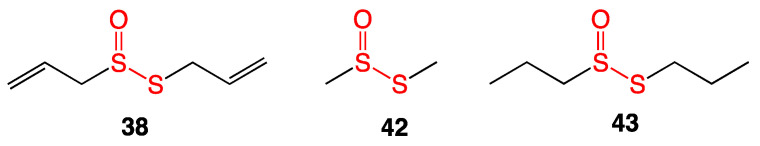
Structures of allicin (**38**)*, S*-methyl methanethiosulfonate (**42**), and *S*-propyl propanethiosulfonate (**43**).

**Figure 19 ijms-24-08659-f019:**
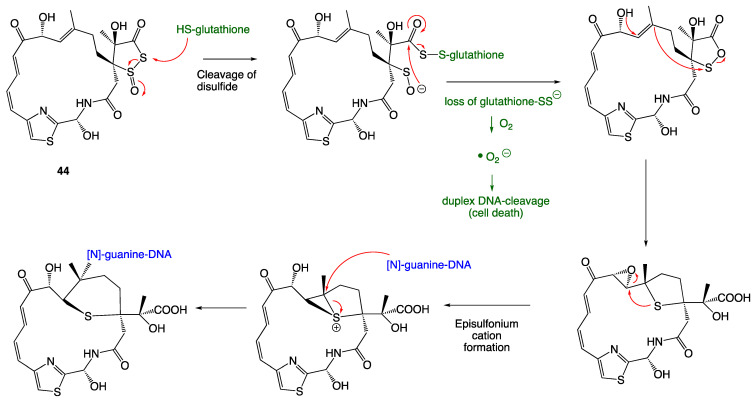
Mechanism of action for formation of leinamycin-DNA adduct [24].

**Figure 20 ijms-24-08659-f020:**
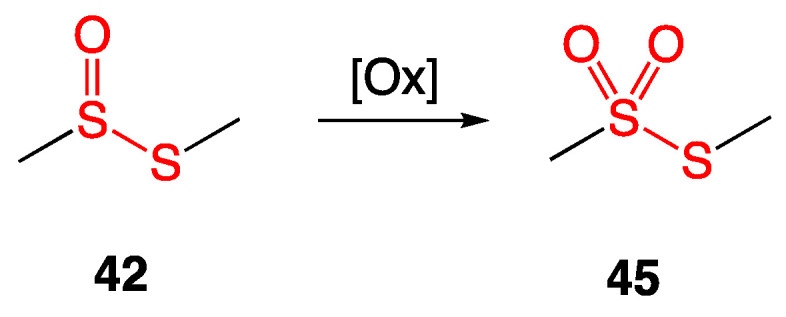
Oxidation of *S*-methyl methanethiosulfinate (**42**) to *S*-methyl methanethiosulfonate (**45**).

**Figure 21 ijms-24-08659-f021:**
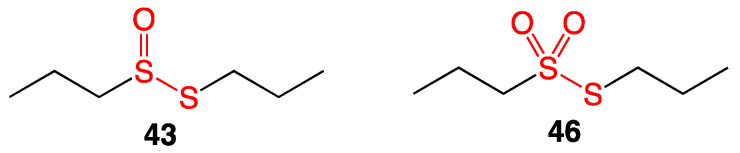
*S*-propyl propanethiosulfinate (**43**) and *S*-propyl propanethiosulfonate (**46**).

**Figure 22 ijms-24-08659-f022:**
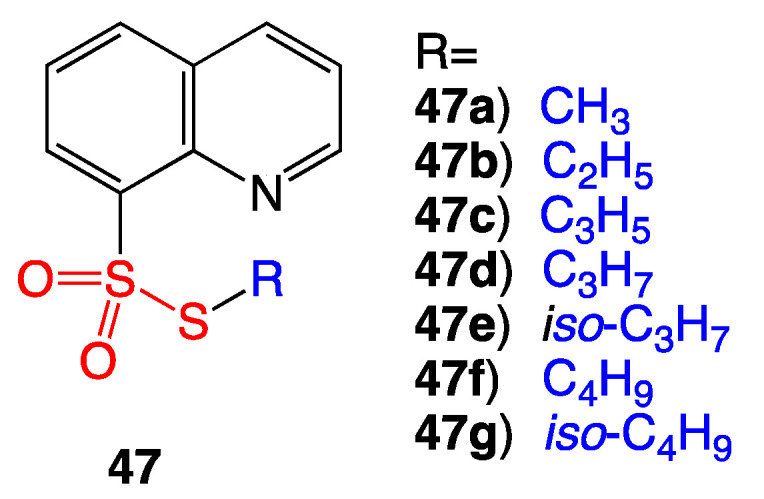
Structure of thiosulfonate quinoline derivatives.

**Figure 23 ijms-24-08659-f023:**
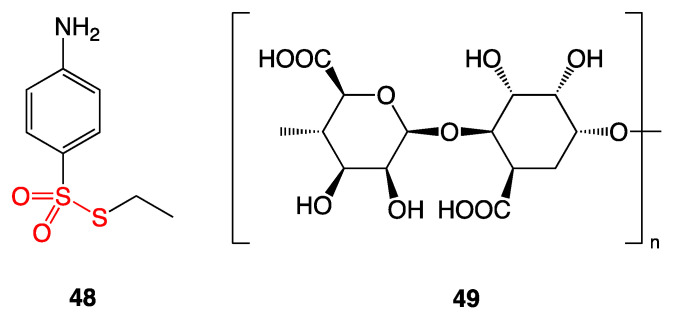
*S*-ethyl para-aminobenzenethiosulfonate (**48**) and biosurfactant rhamnolipid biocomplex PS (**49**).

**Figure 24 ijms-24-08659-f024:**
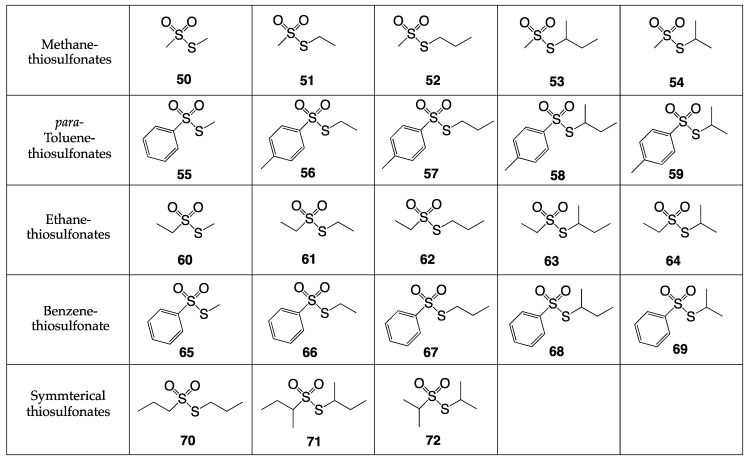
Twenty-three *S*-alkyl thiosulfonate esters synthesized.

**Table 1 ijms-24-08659-t001:** Minimum inhibitory concentrations (MICs) of ajoene [8].

Bacterial Species	MIC (µg/mL)
*Bacillus cereus*	4
*Bacillus subtilis*	4
*Staphylococcus aureus*	16
*Mycobacterium smegmatis*	4
*Mycobacterium pheli*	14
*Streptomyces griseus*	4

Number refers to the minimum inhibitory concentration of compound where bacterial growth was inhibited. MIC values were determined by standard serial dilutions in 24-well plates.

**Table 2 ijms-24-08659-t002:** Zones of inhibition and MIC assays of select aryl–alkyl disulfides.

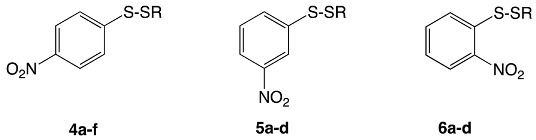				
	**Alkyl**	** *S. aureus* ** **(ATCC25923)**	**MRSA** **(ATCC 43300)**	** *B. anthracis* **
**4a**	Methyl	35 mm 16 µg/mL	23 mm 32 µg/mL	54 mm 1 µg/mL
**4b**	Ethyl	54 mm 1 µg/mL	38 mm 16 µg/mL	67 mm 0.25 µg/mL
**4c**	Isopropyl	85 mm <0.125 µg/mL	75 mm <0.125 µg/mL	71 mm 0.125 µg/mL
**4d**	*sec*-Butyl	68 mm 0.8 µg/mL	54 mm 1.0 µg/mL	62 mm 0.5 µg/mL
**4e**	*n*-Propyl	43 mm 16 µg/mL	38 mm 16 µg/mL	57 mm 1 µg/mL
**4f**	*n*-Butyl	53 mm 1 µg/mL	28 mm 32 µg/mL	52 mm 1 µg/mL
**5a**	Methyl	36 mm 16 µg/mL	15 mm 64 µg/mL	43 mm 8 µg/mL
**5b**	Ethyl	45 mm 8 µg/mL	12 mm 64 µg/mL	40 mm 8 µg/mL
**5c**	Isopropyl	85 mm <0.125 µg/mL	38 mm 16 µg/mL	54 mm 1 µg/mL
**5d**	*sec*-Butyl	45 mm 8 µg/mL	22 mm 32 µg/mL	41 mm 8 µg/mL
**6a**	Methyl	37 mm 16 µg/mL	26 mm 32 µg/mL	31 mm 16 µg/mL
**6b**	Ethyl	40 mm 16 µg/mL	28 mm 32 µg/mL	35 mm 16 µg/mL
**6c**	Isopropyl	31 mm 16 µg/mL	20 mm 32 µg/mL	27 mm 32 µg/mL
**6d**	*sec*-Butyl	24 mm 32 µg/mL	20 mm 32 µg/mL	23 mm 32 µg/mL
Penicillin G (reference)		33 mm 0.03 µg/mL	14 mm 64 µg/mL	nt nt
Vancomycin (reference)		27 mm 0.25 µg/mL	22 mm 0.5 µg/mL	nt nt

The top number associated with each compound and bacterial strain is the average diameter where bacteria did not grow after 24 h of incubation with 20 μg of the compound. The lower number refers to the minimum inhibitory concentration of compound at which bacterial growth was inhibited. MIC values were determined by standard serial dilutions in 24-well plates.

**Table 3 ijms-24-08659-t003:** Comparison of MIC and MBC Values (µg/mL) of selected thiosulfinates [33].

Thiosulfinate	Conc. (µg/mL)	*Escherichia coli K12 Ec*	*Pseudomonas fluorescens Pf*-01	*Pseudomonas syringae* pv. *phaseolicola* 4612 Ps4612	*Micrococcus luteus MI*	*Saccharomyces cerevisiae* BY4742 *Sc*
Dimethyl thiosulfinate	MIC MBC	64 64	16 32	16 16	64 64	16 MFC 16
Diethyl thiosulfinate	MIC MBC	64 64	128 256	8 16	32 32	8 MFC 8
Diallyl thiosulfinate	MIC MBC	32 32	128 256	16 16	32 32	2 MFC 4
Dipropyl thiosulfinate	MIC MBC	32 32	256 256	32 64	32 32	2 MFC 4

The top number refers to the minimum inhibitory concentration of the compound where bacterial growth was inhibited. MIC values were determined by standard serial dilutions in 96-well plates. The bottom number refers to minimum bactericidal concentration of the compound. MBC values were determined after incubation for 24 h. MFC refers to minimum fungal concentration that reduced growth of the fungus *S. cerevisiae*.

**Table 4 ijms-24-08659-t004:** Disk diffusion assay of ointments.

Agent Name		Zones of Growth Inhibition (mm)	
	*Candida albicans*	*Staphylococcus aureus*	*Escherichia coli*
Ointment of **48** and **49**	32.2 ± 0.1	30.0 ± 1.0	23.3 ± 0.8
Clotrimazole ointment	26.5 ± 0.2	18.6 ± 0.4	20.3 ± 0.7
Econazole gel	--	17.5 ± 0.5	16.4 ± 0.6
Nystatin ointment	24.2 ± 0.5	0	0
Gentamicin ointment	13.5 ± 0.2	26.3 ± 0.7	25.5 ± 0.5

**Table 5 ijms-24-08659-t005:** MIC data of *S*-alkyl thiosulfonate esters **50**–**72**.

	MRSA COL	VISA MU50	VRSA-1	VRE HF50104	KP 700603	AB5075-UW	PA 15442	EC13047
**50**	>64	>64	>64	>64	>64	>64	>64	>64
**51**	>64	>64	>64	>64	>64	>64	>64	>64
**52**	64	64	64	64	64	64	>64	64
**53**	32/16	32	32	64	>64	64	>64	64
**54**	>64	>64	>64	>64	>64	>64	>64	>64
**55**	64	64	64	64	>64	64	>64	>64
**56**	64/>64	64	64	>64	>64	>64	>64	>64
**57**	64	64	64	64	>64	>64	>64	>64
**58**	16/8	16	32/16	64	>64	64	>64	>64
**59**	64	64	64	>64	>64	>64	>64	>64
**60**	64	64	64	64	64	64	64	64
**61**	64/>64	64	64	64	>64	64	>64	>64
**62**	16	16	16	32	64	32	>64	32
**63**	16/8	8	8	32	>64	64	>64	64
**64**	8	8	8	32	64	32	>64	32
**65**	32	>64	64	32	>64	64	64	64
**66**	32	>64	>64	32	>64	64	>64	64
**67**	>64	>64	>64	16	>64	>64	>64	>64
**68**	16	>64	32	8	>64	>64	>64	>64
**69**	8	16/4	16	8	>64	>64	>64	>64
**70**	8	4	2	32	>64	>64	>64	>64
**71**	8	4	2	32	>64	>64	>64	>64
**72**	>64	>64	>64	>64	>64	>64	>64	>64
COLISTIN	>64	>64	>64	16	1	≤1	1	>64
VANCOMYCIN	2	8	>64	64	>64	64	>64	>64

The number refers to the minimum inhibitory concentration (μg/mL) of compound where bacterial growth was inhibited. MIC values were determined by standard serial dilutions in 24-well plates in triplicate assays.

**Table 6 ijms-24-08659-t006:** MIC data of most bioactive *S*-alkyl thiosulfonate esters.

	MRSA COL	VISA MU50	VRSA-1	VRE HF50104	KP 700603	AB5075-UW	PA 15442	EC13047
**58**	16/8	16	32/16	64	>64	64	>64	>64
**63**	16/8	8	8	32	>64	64	>64	64
**64**	8	8	8	32	64	32	>64	32
**68**	16	>64	32	8	>64	>64	>64	>64
**69**	8	16/4	16	8	>64	>64	>64	>64
**70**	8	4	2	32	>64	>64	>64	>64
**71**	8	4	2	32	>64	>64	>64	>64
Colistin	>64	>64	>64	16	1	≤1	1	>64
Vancomycin	2	8	>64	64	>64	64	>64	>64

The number refers to the minimum inhibitory concentration (μg/mL) of compound where bacterial growth was inhibited. MIC values were determined by standard serial dilutions in 24-well plates in triplicate assays.

## Data Availability

Please contact authors directly for questions or requests for supporting data on compounds **50**–**72**.

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
