# Peer review of "Applications and Opportunities in Using Disulfides, Thiosulfinates, and Thiosulfonates as Antibacterials"

_ijms, 2023, doi:10.3390/ijms24108659_

Round 1
Reviewer 1 Report
The manuscript is well organized and written, I have no comment on that. Some liberties in the formatting of tables are the responsibility of the Editor, who will be able to make a decision on this issue. It is also necessary to check the correct formatting of the text: "text [ref]." instead of "text.[ref]"
I am not a native speaker, however, in my opinion, the manuscript is well written and understandable.
Author Response
We sincerely appreciate the comments of referee, and note that some of the errors appearing in the tables and even in the text or figures came from the editorial reformatting of the original Word file. We have redone and fixed these throughout the text, figures and schemes, in the reformatted version, which has all the corrections and additions color-highlighted.
Reviewer 2 Report
The review manuscript entitled "Applications and opportunities in using disulfides, thiosulfinates, and thiosulfonates as antibacterials" by Blume et al involved the current interesting bibliographical research of the organosulfur compounds with antibacterial properties. Moreover, the research group has experience in this specific subject.
The introduction is according to the developed topic of the manuscript.
Also, the manuscript is clear, organize, and easy to follow according to the goal of the authors.
In addition, the information they described is supported with figures/images/schemes that summarize all the data.
Some questions to considering in order to complete the information:
- Did the authors check the current state of art of the patents related to the molecules they reported? It is important to analyze this data to improve the article for the readers.
- Please check the following references to complete the reported information:
Molecules 2022, 27(20), 6900; https://doi.org/10.3390/molecules27206900
Pharmaceuticals 2021, 14(1), 21; https://doi.org/10.3390/ph14010021
However, some typo mistakes should be corrected such as (highlighted in yellow, pdf attached):
· Please remember that the unit and the number (e.g. 20 nM – 30 °C – 10 min – 2 h – 2 Å) should have a blank space between them. Please check them all throughout the manuscript.
·
- S-alkyl; pyridine-N-oxide; etc.: S and N should be in italics. Please check them all throughout the manuscript
- The names of the bacterial species should be in italics: S. aureus as an example, please check them all throughout the manuscript.
- It is important to improve the quality of the scheme figures containing molecules because they look blurry.
- Please homologate the style for all the tables included in this manuscript (size, colour, etc.)
Furthermore, I encourage the authors to include a table summarizing all the information reported in the present manuscript.
Finally, I would like to invite the authors to add the abbreviation list of words at the end of this manuscript.

Author Response
We sincerely appreciate the comments of referee, and note that some of the errors appearing in the tables and even in the text or figures came from the editorial reformatting of the original Word file. We have redone and fixed these throughout the text, figures and schemes, in the reformatted version, which has all the corrections and additions color-highlighted. Including the italicizing of "S", "N" and "bis" and all the Latin names of the bacterial strains, etc.. We have added the references noted by the reviewer, with further discussion where needed. Thanks! We note that in conducting the literature review we examined all scientific and patent publications available to us, as completely as we could. Only those reporting on the biological effects of the sulfur compounds were including in the references and discussion. For the tables, we have indicated the concentrations for the MIC and MBC values (as mg/mL). We thought to also summarize the results and the abbreviations in tables, as suggested, but decided against it because it would require significant more space and the abbreviations are generally defined where they are used int he text. So we are happy to put this together if this is still deemed useful, but the referee or the editor. We only are interested in providing concise but useful information about this topic to the readers. Finally, we have put in additional commentary regarding some future opportunities for applying these types of molecules for antibacterials development, at the very end of the paper.
Reviewer 3 Report
Sulfur-containing molecules have a long history of use as antibacterial agents, both in natural products and commercial antibiotics. A review focuses on disulfides, thiosulfinates, and thiosul-fonates as potential antibacterial compounds and highlights opportunities for future development. The authors need to address the following concerns before it is considered to be published.
1. Please add references in the second paragraph of the introduction.
2. The chemical structures are blurry in Figure 1 to 7, 9, 12, 14, 16 to 18, and 20 to 23, but some are clear, like Figure 11 and 19. Please update accordingly.
3. Please emphasize more future development of those compounds.
Author Response
We sincerely appreciate the comments of referee, and note that some of the errors appearing in the tables and even in the text or figures (such as blurriness, which we do not understand how that could happen) came from the editorial reformatting of the original Word file. We have redone and fixed these throughout the text, figures and schemes, in the reformatted version, which has all the corrections and additions color-highlighted. We have added references in the second paragraph of the introduction, as requested, and additional commentary regarding some future opportunities for applying these types of molecules for antibacterials development, at the very end of the paper.